# Analysis of Spatio-Temporal Dynamics of Chinese Inland Water Clarity at Multiple Spatial Scales between 1984 and 2018

**Hui Tao** [1,2] , **Kaishan Song** [1,3,*], **Ge Liu** [1], **Qiang Wang** [1,2], **Zhidan Wen** [1] , **Junbin Hou** [1], **Yingxin Shang** [1] **and Sijia Li** [1]

[1] Northeast Institute of Geography and Agroecology, Chinese Academy of Sciences, Changchun 130102, China
[2] School of Resources and Environment, University of Chinese Academy of Sciences, Beijing 100049, China
[3] College of Urban Research and Planning, Liaocheng University, Liaocheng 252000, China
* Correspondence: songkaishan@iga.ac.cn

**Abstract:** Water clarity (Secchi disk depth, SDD) provides a sensitive tool to examine the spatial pattern and historical trend in lakes' trophic status. However, this metric has been insufficiently explored despite the availability of remotely-sensed data. Based on the published SDD datasets derived from Landsat images, we analyzed the spatial and inter-annual variations in water clarity and examined the impact of natural and anthropogenic factors on these trends at multiple scales, i.e., five lake regions, provinces, and watersheds. Lake clarity was lowest in Northeast ($0.60 \pm 0.09$ m) and East China ($1.23 \pm 0.17$ m) and highest in the Tibet Plateau ($3.32 \pm 0.38$ m). Over the past 35 years, we found a significant trend of increased SDD in 18 (out of 32) provinces (only Yunnan province exhibited a significant decreasing trend) and in 77 (out of 155) watersheds (only 5 watersheds showed a significant decreasing trend). Lakes in eastern-northeastern China exhibited a higher probability of decreasing trend, while the trend was inverse for lakes in the Tibet-Qinghai region. The results of water clarity interannual change trends showed they were closely related to the spatial scale of analysis. At the watershed level, these trends were mainly driven by anthropogenic factors, with night-time brightness (13.84%), agricultural fertilizer use (11.17%), and wastewater (9.64%) being the most important. Natural factors (temperature, wind, and NDVI) explained about 18.2% of the SDD variance. Our findings for the SDD spatio-temporal trend provide valuable information for guiding water protection management policy-making and reinforcement in China.

**Keywords:** Landsat images; climate change; anthropogenic activities; water clarity; multiple scales



## 1. Introduction

Lakes and reservoirs (herein lakes) play crucial roles in the aquatic environment for wildlife and serve as freshwater water sources for drinking, industrial, and agricultural uses [1–3]. No doubt, China has made huge achievements, especially in socio-economic development, since the "Reform and Opening-up" policy was initiated in 1978. However, China also has been facing increasingly severe water pollution, scarcity, and security problems ever since [4]. Agricultural nonpoint pollution, industrial production-related pollution, population growth, and urban expansion have exerted increasing stress on water quality [4,5]. Eutrophication is the most severe problem with respect to water quality, and great efforts have been devoted to controlling or reducing trophic water levels [6–8]. At the same time, China also invests a huge amount of money in environmental improvement, which has exerted a strong impact on the environment in the last 20 years [6,9]. Further, afforestation and returning grassland or wetlands from agriculture projects have also been reinforced in the past three decades, which have turned China greener [10].

Across the country, a limited number of stations have been deployed over different sections of inland waters to monitor water quality [11]. As for water resource managers in China, they need suitable assessment tools to monitor inland water quality over

time [12]. Water clarity or transparency is a comprehensive proxy for evaluating the water trophic state [13], which is closely linked with the presence of suspended sediment, planktonic algae, zooplankton, and colored dissolved organic matter (CDOM) in the water column [14–16]. Water clarity is commonly measured by Secchi disk depth (SDD) which is relatively easy to operate and, thus, is a very practical measure to monitor water quality [17,18]. Nevertheless, these traditional approaches are limited in their suitability for monitoring water bodies with large surface areas due to the dynamic nature and lakes in large geographic regions [19,20]. Though the operation of SDD measurements is easy, lakes in remote areas, large geographic regions, or without aquatic vehicles make it impossible to collect clarity data [21,22]. Optically active constituents (OACs), i.e., phytoplankton, non-algal particle, and CDOM, govern water clarity and are the major components that determine water-leaving radiance, which can be detected by remote sensing sensors [23,24]. Remote sensing has been widely used for monitoring the spatio-temporal dynamics of SDD at regional, national and global scales [21,25–27]. The Landsat images can be used to monitor the spatio-temporal variation of SDD and track the long-term trend in the past 35 years with its free-of-charge images. Such an analysis can provide valuable information for evaluating water eutrophication status, which can be used for water resource planning and management or attempting to decrease eutrophic levels [27–29], yet the option of applying long-term archived Landsat images has not been fully explored.

The spatial distribution and temporal variation of SDD can be influenced by natural factors and anthropogenic activities [25,30]. In recent years, a few researchers have examined the driving factors for SDD interannual changes across China. Liu et al. (2020) paid more attention to the influence of natural elements on SDD variations across China, i.e., wind, water depth, temperature, NDVI, precipitation, and basin slope, and only chose one anthropogenic element (population). Wang et al. (2020) selected four natural factors (precipitation, temperature, lake depth, lake altitude) and two anthropogenic factors (population, GDP) to explore the linkage between these factors and SDD variations. The above studies mainly focused on the large lakes and reservoirs (area > 10 km$^2$), and the anthropogenic factors they selected represented the indirect effects of human activities on water quality. In this study, the combined effect of climatic (temperature, wind, precipitation, NDVI) and socio-economic factors (agriculture fertilizer use, wastewater discharge, NTL) on the temporal and regional variability of lake SDD (area > 1 ha) have not been systematically examined. The overall purpose was to explore the natural and anthropogenic factors driving SDD variations based on published SDD datasets (1984–2018) derived from Landsat images [31]. Specifically, the objectives were to: (1) examine the inter-annual variability of SDD at multiple scales, specifically, five lake regions, provinces, and watersheds; (2) investigate the correlation analysis and relative contribution of natural and anthropogenic factors on the temporal trends of lake SDD at each of these scales through hierarchical partitioning for canonical analysis.

## 2. Materials and Methods

### 2.1. Study Area

According to the regional characteristics of geography and climate, our study regions across China have been grouped into five lake regions [32], i.e., the Inner Mongolia–Xinjiang lake region (MXR), Tibetan–Qinghai Plateau lake region (TQR), Northeastern lake region (NLR), Yungui Plateau lake region (YGR), and Eastern lake region (ELR) (Figure 1a). The MXR and TQR are located in arid or semi-arid climates and have higher evaporation and lower annual precipitation and temperature than the other three lake regions. Additionally, the lakes situated at high altitudes in the plateau region are less affected by human activities and generally display better ecological conditions than lakes in the other regions [33]. The NLR, YGR, and ELR are influenced by the Asian monsoon climate, and these lakes are frequently influenced by anthropogenic activities [9,28]. During the period 1984–2018, the characteristics of climate in the five lake regions exists discrepancy in space (Figure 1b–d), i.e., annual mean precipitation is in the descending order: ELR > YGR > NLR > TQR > MXR, an-

nual mean wind speed is: NLR > MXR > ELR > TQR > YGR, annual mean temperature is: ELR > YGR > ELR > TQR > YGR. The lakes with SDD records of more than 10 years obtained from Tao et al. (2022) include natural lakes and artificial lakes (also named reservoirs), with sizes ranging from 1 ha to 4,483 km$^2$ (Qinghai Lake, 2018). The maximum mean depth of the lake, named Changbaishan Heavenly Lake, is up to 204 m. The trophic status of lakes in China varies from oligotrophic to mesotrophic to eutrophic, among which the lakes releasing signals of eutrophic are mainly distributed in the ELR region [15].

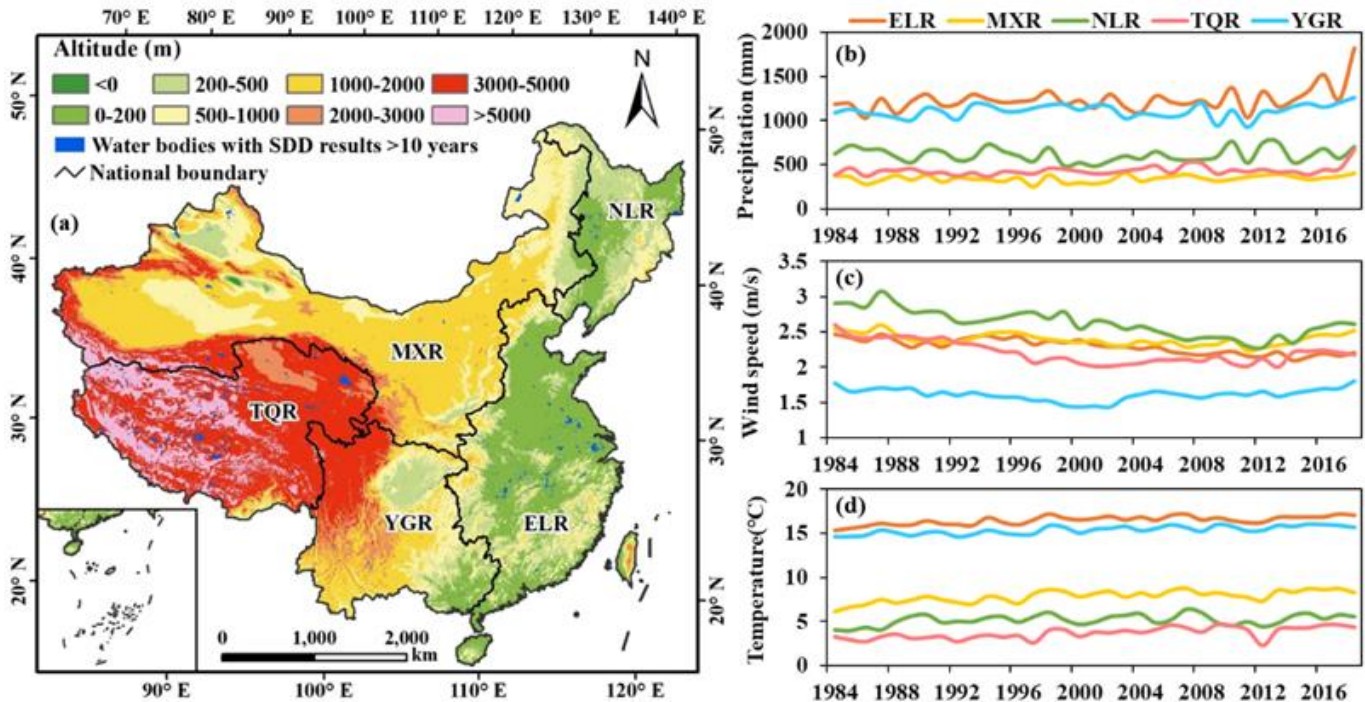

**Figure 1.** Examined lakes distributed in five lake regions across China (**a**) and annual mean meteorological data in five lake regions during 1984–2018, i.e., precipitation (**b**), wind speed (**c**), and temperature (**d**). Note that the data on water bodies were sourced from Tao et al. (2022), and the meteorological data derived from the National Meteorological Information Center.

*2.2. Collection of Climatic and Socio-Economic Data*

To help with the interpretation of results, we used three types of data in this study: (1) climatic and physiographic data, including vegetation cover (as expressed by normalized difference vegetation index or NDVI), annual average temperature, wind speed, and precipitation; (2) data on anthropogenic activities, including agricultural fertilizer use, wastewater discharge (industrial and sanitary sewage), and night-time light brightness (NTL); (3) other ancillary data. More details about the various data sources are provided in Table S1.

NDVI is a proxy for vegetation growth in a drainage basin and can affect soil erosion and sediment discharge into lakes [34]. In this study, using NDVI as a surrogate for vegetation cover, we downloaded AVHRR GIMMS NDVI data at 1/12-degree spatial resolution from 1984 to 2015 (https://ecocast.arc.nasa.gov/data/pub/gimms/, accessed on 10 October 2022), and MODIS NDVI MOD13C1 products at 5.6 km spatial resolution from 2016 to 2018 (https://modis.gdfc.nasa.gov/, accessed on 10 October 2022). For this research, both of the NDVI datasets were resampled to 8-km spatial resolution using the maximum value compositing procedure to minimize the effect of cloud contamination (mean annual NDVI are presented in Figure S1a) [35]. In order to quantify the impact of climatic factors on SDD, we downloaded yearly average temperature, wind speed, and precipitation from the National Meteorological Information Center (http://data.cma.cn/, accessed on 10 October 2022). Data were obtained from 613 stations distributed across China.

The total amount of fertilizer used for agriculture and wastewater discharge (sum of industrial wastewater and domestic wastewater) was obtained from the Provincial Statistical Yearbook (http://tongji.cnki.net/, accessed on 10 October 2022). The two types of data were related to water quality change as sources of TN, TP, and nutrients [9]. The nighttime light brightness (NTL) data were derived from products of the operational Line-scan System of Defense Meteorology Satellite Program (DMSP-OLS) of the National Oceanic and Atmospheric Administration's (NOAA's) National Geographic Data Center (NGDC) across continental China during 1992–2013 (https://ngdc.noaa.gov/eog/download.html, accessed on 10 October 2022), which has a strong correlation with human activities [36]. We chose stable light data, one of the three kinds of annual average data covering cities, townships, and other lasting light emissions, and background noise was removed. Because the original NTL data has a problem of non-continuity, non-comparability, and saturation, according to the invariant region method of Liu et al. (2012), we resolved the first two questions, and the last one was worked out through the approach proposed by Wu et al. (2013), which consists of selecting a suitable reference NTL image with no sensor saturation. Following these steps, data were projected with the Lambert Azimuthal Equal Area projection and resampled to a pixel size of 1 km (mean annual NTL is presented in Figure S1b).

### 2.3. Methods

### 2.3.1. Analyzing the Dynamics of SDD

Based on the published annual mean SDD datasets of China between 1984 and 2018, there were 10,814 lakes remaining for analyzing the interannual dynamics of SDD with results of each lake for more than 10 years [31]. The datasets were generated by using empirical algorithms (red/blue band ratio) based on the Landsat top-of-atmosphere reflectance product with Google Earth Engine, where $R^2 = 0.79$, RMSE = 100.30 cm, relative RMSE = 61.90%, and MAE = 57.70cm. Before analyzing the dynamics of SDD, the results of annual mean SDD needed to be obtained at three scales across China, i.e., five lake regions, provinces, and watersheds. A total of 34 provinces (Figure S2a) and 161 watersheds (Figure S2b) in China were included when calculating SDD results. Due to data deficiency (Taiwan) or a limited number of lakes (e.g., Hong Kong and Macao), Taiwan, Hong Kong, and Macao were excluded from the analysis when assessing for the relative contribution of driving factors. To define watershed boundaries, we first obtained maps of secondary and tertiary basins, and these were complemented with relational data about the basic hydrological characteristics of each watershed. Maps were then georeferenced using available tools in the ArcGIS 10.3 software package. After that, for each lake region, province, and watershed, the mean, standard deviation, and change trend of SDD were calculated. The interannual change trend of lake SDD was obtained using linear regression analysis using IBM SPSS Statistics 22. Based on the significance level (5%) and slope from the linear regression model between SDD values and year, there were three interannual change trends: significant increasing (slope > 0 and $p < 0.05$), significant decreasing (slope < 0 and $p < 0.05$), and non-significant change ($p > 0.05$).

### 2.3.2. Quantizing the Driving Factors

Generally, we have trouble directly obtaining datasets of wastewater discharge and agricultural fertilizer use within the extent of watersheds, which were recorded at the province level. Therefore, before analyzing the correlation and contribution between driving factors and lake SDD, the two types of datasets should be derived at the watershed level. We referenced the province-to-watershed conversion methods to obtain these two datasets, and a detailed description can be seen from the study by Ma et al. (2020) and Fang et al. (2022) (Figure S3). To obtain province-level and watershed-level NTL data based on vector boundaries of provinces and watersheds on the platform of ArcGIS 10.3, interpolation and extrapolation using linear regression were applied to obtain continuous data from 1984 to 2018. As for the meteorological data, Stations within each province and watershed

boundary were selected to calculate the mean values of climatic factors. If a region did not contain any stations, the closest station was selected to estimate the relative mean values for that region (Figure S4). The datasets of the NDVI at the province and watershed scales were directly obtained from the rater datasets according to the corresponding boundaries.

### 2.3.3. Statistical Analysis

Based on the province and watershed scale datasets of driving factors prepared, when a significant trend (increasing or decreasing) in SDD is detected, correlation analysis was conducted (at the province or watershed scale) to examine linkages with anthropogenic and natural factors. Statistical significance was determined at the 1% or 5% level, and analysis was conducted with the IBM SPSS software. Finally, we conducted a hierarchical partitioning for canonical analysis (HPCA) using the RStudio 3.6.2 statistical software package (named "rdacca.hp") to estimate the relative contribution of the seven factors considered as potential driving forces of inter-annual variation in lakes' SDD. A notable advantage of this method is that it eliminates the effect of multi-collinearity among explanatory variables [37]. Additionally, the method of HPCA could standardize each variable in the process of contribution analysis, which is conducive to analyzing the relative importance of different variables.

## 3. Results

### 3.1. Spatial Pattern in Lake Water Clarity

Water clarity demonstrated remarkable spatial variation across the lake regions of China, with lakes in the mountain regions generally exhibiting higher SDD than those situated in the flat plain regions (Figure 2a). Among the five lake regions, lakes in the NLR exhibited the lowest SDD (mean: $0.60 \pm 0.09$ m), followed by the ER (mean: $1.23 \pm 0.17$ m). The MXR showed intermediate SDD values (mean: $1.63 \pm 0.38$ m), but lakes located in the high mountain areas in the mid-south and northwest sections of that region exhibited higher SDD. Lakes in the YGR exhibited relatively higher SDD (mean: $2.35 \pm 0.21$ m), likely due to their location on the Yungui Plateau, where dense forests prevail. Lakes in the TQR had the clearest water (mean SDD: $3.32 \pm 0.38$ m) but also exhibited the widest variation in SDD (range: 0.13–12.70 m). The TQR includes large, deep, and clear alpine lakes on the Tibet Plateau, as well as multiple small and shallow lakes that tend to be relatively turbid.

Variation in lake SDD was examined at the provincial scale (Figure 3a). Lake SDD was lowest in the Heilong Jiang, Jilin, and Inner Mongolia provinces (in NLR) and in the Jiangsu, Anhui, and Jiangxi provinces (in ER). These turbid lakes exhibited a narrow range (<1 m) in SDD and small standard deviation (S.D). In contrast, for provinces in mountainous areas of south and southwest China (Chongqing, Sichuan, Yunnan, Zhejiang) and in the TQR region (Xinjiang), lake SDD was the highest (>2 m), but the variation in SDD was widest (hence, large standard deviation). Lakes in the other provinces exhibited intermediate SDD (1–2 m) and also intermediate variation in S.D.

SDD spatial pattern at the watershed scale generally resembled that at the provincial scale (Figure 3b). As an illustration of the effect of land use and land cover (Figure S5), lakes in the mountainous region and on the Tibetan Plateau (located in watersheds with dense forest cover) generally showed high SDD and also larger S.D. values. Conversely, lakes in the flat plain regions, e.g., the Songnen Plain, the Yangtze Plain, and also lakes situated in north and northwest grassland or desert landscapes with semi-arid or arid climates, displayed lower SDD with less inter-annual variation (S.D. < 0.5).

### 3.2. Temporal Trend in Lake Water Clarity

With the exception of the TQR, results for the other four lake regions indicated a significant ($p < 0.05$) increasing trend in SDD during the study period (Figure S6). We further examined the data at the provincial and watershed scales to examine the impact of the scale of analysis on the temporal trend in lake SDD. At the provincial scale, lake clarity in 18 provinces showed a significant increasing trend, with 1 province showing

a significant decreasing trend (e.g., Yunnan Province) in the past 35 years (Figure 2b). Specifically, according to Figure 3a, c, the number of provinces with mean SDD values that fell into the levels of 0 to 0.5 m, 0.5 to 1 m, 1.0 to 1.5 m, 1.5 to 2 m, 2.0 to 2.5 m, and >2.5 m represented 2 (5.88%), 5 (14.71%), 9 (26.47%), 3 (8.82%), 5 (14.71%), and 8 (23.53%) among all 32 provinces (Hong Kong and Macao were merged into Guangdong province since there very limited water bodies in these two special administrative units), respectively. Among these categories of SDD, the number of provinces with SDD increase was 1, 3, 7, 0, 3, and 5, respectively.

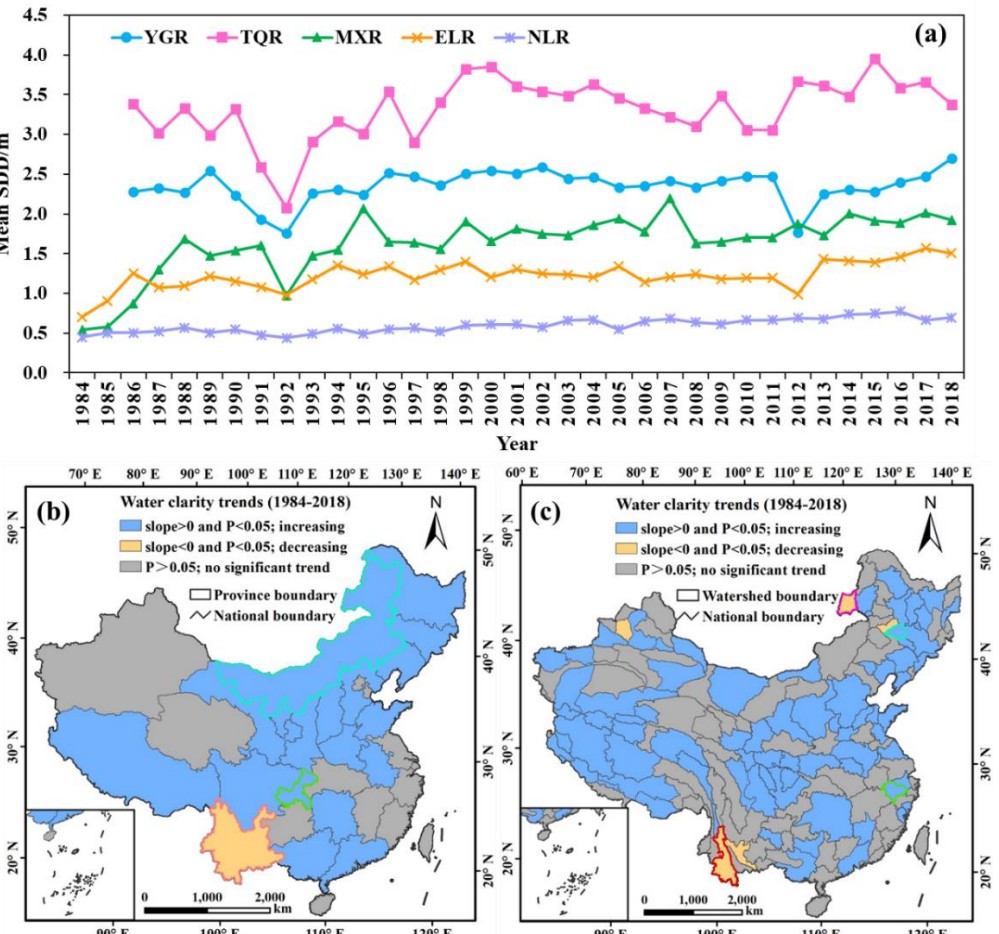

**Figure 2.** Interannual trends (1984–2018) in lakes SDD in five lake regions of China (**a**). Significant decreasing (slope < 0 and *p* < 0.05), significant increasing (slope > 0 and *p* < 0.05), and non-significant trends in SDD are presented for different provinces (**b**) and at the watershed scale (**c**). The green and purple boundaries depict, respectively, regions of maximum increasing and decreasing trends in SDD; the bright green and red boundaries represent, respectively, areas of minimum increasing and decreasing trends. The significance of slope is determined by *t*-test at the 5% significance level.

At the watershed scale, the country was divided into 161 watersheds, but 6 watersheds were excluded since they include no water body with 10 years of SDD records; thus, only 155 watersheds were further analyzed (Figure 2c; Figure 3b). Altogether, 77 watersheds showed a significant increasing trend (Figure 2c), with the Qiandao Lake watershed exhibiting the largest increasing trend (i.e., slope). Lake SDD in 5 watersheds showed a decreasing trend, and no clear inter-annual trend was observed in the other watersheds (Figure 2c). Using the same categories of annual mean SDD values that were employed at the province level (i.e., 0 to 0.5 m, 0.5 to 1 m, 1 to 1.5 m, 1.5 to 2 m, 2 to 2.5 m, and >2.5 m), the number of watersheds in each category was 14 (8.70%), 28 (17.39%), 39 (24.22%), 17 (10.56%), 7 (4.35%), and 50 (31.06%) among all 155 watersheds, respectively. The number of watersheds with a

significant increase in SDD was 10, 13, 26, 7, 3, and 23, respectively (Figure 3b,d). Based on the above results, it could be seen that SDD interannual change trends depended on the spatial scale of analysis, a result that needs to be considered when analyzing for driving forces.

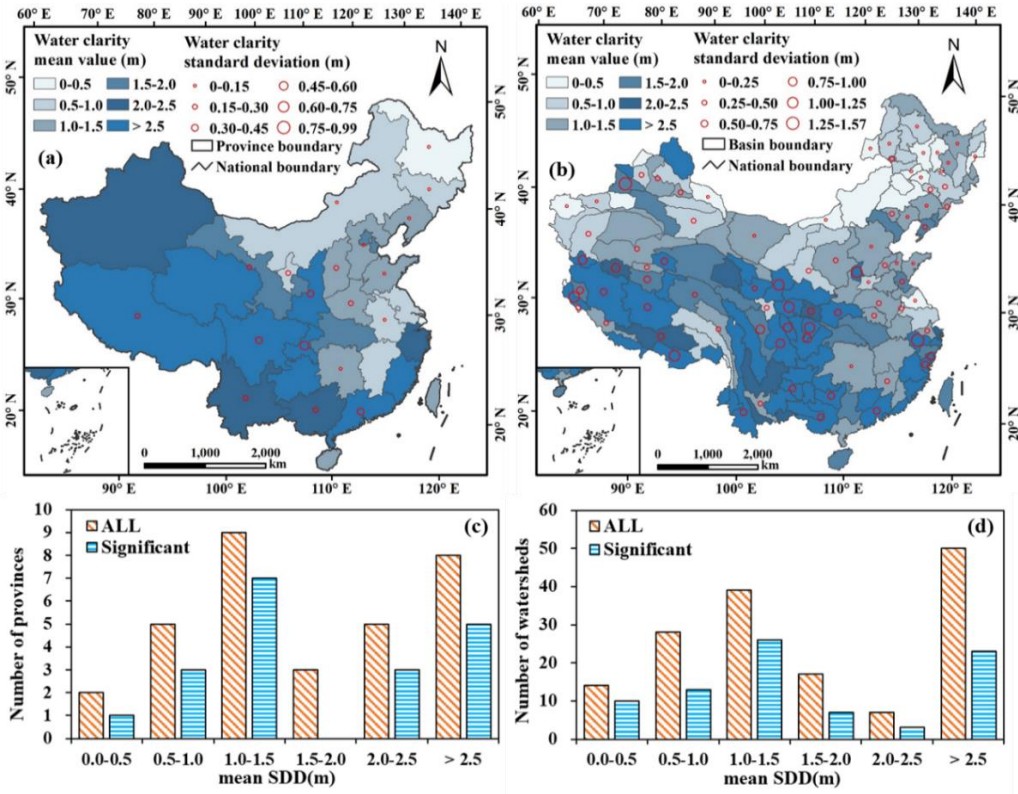

**Figure 3.** Average and standard deviation (red circles) of lake SDD during 1984–2018 by provinces (**a**) and by watersheds (**b**). Standard deviation data are presented only for the 19 provinces (**a**) and 82 watersheds (**b**) that had shown significant (*p* < 0.05) trends in interannual changes in mean SDD. The histograms showed the distribution of lake SDD among provinces (**c**) and watersheds (**d**). Histograms (**c**,**d**) are drawn for all lakes (brown hatched bars) and for the subset of lakes where a significant trend in SDD was detected.

## 4. Discussion

### 4.1. Natural Versus Anthropogenic Factors

It is widely accepted that lake SDD can be affected by OACs, and the abundance of these compounds in the water column can be influenced by natural factors and anthropogenic forces [28,38]. For this study, a set of seven driving factors (natural and anthropogenic) was considered, and the correlation between these factors and changing trends in SDD was examined across 19 provinces and 82 watersheds at the 5% and 1% significance levels (Figure 4). As for natural factors at the province level, lakes from 4 (*p* < 0.01) and 6 (0.01 < *p* < 0.05) provinces demonstrated a significant association between NDVI (normalized difference vegetation index). As for temperature, 11 (*p* < 0.01) and 3 (0.01 < *p* < 0.05) of the 19 provinces revealed a significant association with the SDD changing trend. Wind speed also exerted a strong effect on SDD, with a significant effect on changing SDD trends detected in 4 (at 1% level) and 3 provinces (at 5% level). Precipitation only showed a strong association with changing trends in lake SDD at the 5% level. Precipitation affects surface runoff and the delivery of materials (nutrients, suspended particles) into lakes, while wind speed causes sediment resuspension in the water column, particularly in shallow water bodies [16]. Vegetation cover (NDVI) has a great contribution to soil conservation, which can alleviate erosion and reduce the transportation of soil particles into lakes [39]; thus, we used NDVI as a surrogate for vegetation cover and examined its association with change

in lake SDD. Temperature increase may enhance algal growth, coupled with elevated TP and TN loadings, which might ultimately affect SDD [8,40,41]. Further, temperature can also promote vegetation growth in regions where precipitation is not limiting [10,42]. In addition, the lake area and water depth, not being considered in this study due to the lack of data, are important driving factors for SDD spatio-temporal variation at the region scale [19,39]. Lake expansion and increasing water depth could weaken sediment resuspension, resulting in clearer water quality.

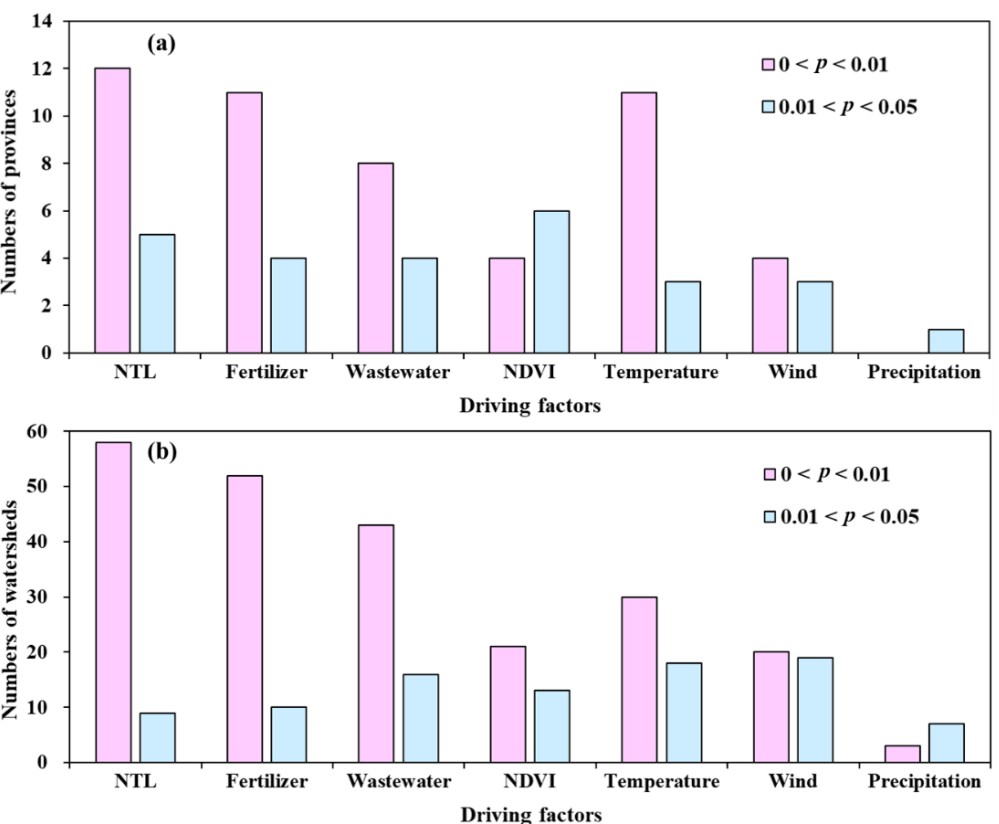

**Figure 4.** Histograms showing the number of provinces (**a**) or watersheds (**b**) where each of the listed natural and socio-economic driving factors had a significant effect on SDD. Histograms are shown for analysis conducted at the 5% and 1% significance levels.

Compared to natural factors at the province scale, anthropogenic factors exerted a much stronger impact on lake SDD (Figure 4a and Figure S7a,b). NTL (night-time light brightness) had a strong association with changing trends in SDD, with lakes from 12 and 5 provinces showing significant associations at the 0.01 and 0.05 levels, respectively. It is well understood that NTL is a comprehensive proxy for socio-economic development and can have strong links with GDP, population size, impervious surface area, and degree of urbanization [10,43]. Thus, it is reasonable that NTL was found to be highly associated with a change in SDD and able to explain a high percentage of the variance in SDD changing trend. Similarly, fertilizer use (11 provinces at $p < 0.01$, 4 provinces at $0.01 < p < 0.05$) and wastewater (8 provinces at $p < 0.01$, 4 provinces at $0.01 < p < 0.05$) also exerted a strong impact on SDD changing trends.

At the watershed scale, the impact of natural and anthropogenic factors on SDD mirrored the patterns observed at the provincial scale (Figure 4b and Figure S7c,d), e.g., the stronger impact of anthropogenic factors than natural factors on lake SDD (Figure 4). Of the 82 watersheds with a significant changing trend in SDD, precipitation had an effect in only 3 watersheds (at $p < 0.01$ level) and 7 watersheds ($0.01 < p < 0.05$ level). Temperature, NDVI, and wind speed had a relatively strong impact on the change in SDD (Figure 4b and Figure S7c,d). Among the anthropogenic factors, NTL had the strongest

impact on change in SDD, with significant impact detected in 58 and 9 watersheds at 1% and 5% levels, respectively. Both fertilizer use and wastewater discharge had a strong impact on SDD change, with a significant impact detected in 52 and 43 watersheds at $p < 0.01$ level. Wastewater (sanitary sewage and industrial sewage) generally contains a high amount of CDOM, which strongly affects light attenuation and SDD [44]. Further, sewage discharge delivers a high amount of TN and TP to water bodies, which stimulates algal growth and ultimately affects lake SDD [13,29,45].

### 4.2. Relative Contribution of the Driving Factors

To quantify the relative contribution of the seven natural and anthropogenic factors on temporal trends in SDD over the past 35 years, HPCA was carried out. At the province level, the contribution of several factors was found to vary among the different provinces (Figure 5a and Figure S8a). The driving factors can explain >70% of the variance in SDD in Heilongjiang and Henan provinces but only <25% in Guangdong and Shandong provinces. Among the anthropogenic factors, NTL contributed to a large proportion of the SDD variance, ranging from 3.34% in Guangdong province to 21.05% in Heilongjiang province. NTL is a comprehensive variable and, among other things, reflects socio-economic development, population, and construction investment [36,46]. As such, NTL can co-vary with wastewater discharge. Fertilizers and wastewater also played an important role in controlling SDD change in some provinces, contributing >10% of the variance in provinces such as Anhui, Henan, and Inner Mongolia. NDVI explained 5.82% of the variation in lake SDD, particularly in the provinces where significant changes in NDVI occurred during the past 35 years (Figure 5a). NDVI is a comprehensive index, reflecting not only weather-related (temperature, precipitation) parameters that stimulate vegetation growth but also the effect of ecological restoration projects [10,47]. Thus, the impact of NDVI may include the contribution of temperature and precipitation, as well as the effect of management of other beneficial human interventions [10,48]. Surprisingly, wind speed and precipitation explained a small proportion of the variance (5.51% and 2.46%, respectively) in lake SDD. Altogether, in 9 provinces (out of 19 provinces), anthropogenic factors explained more of the variance in lake SDD than natural driving factors ($27.50 \pm 13.44\%$ and $22.28 \pm 7.93\%$, respectively).

At the watershed scale, the seven natural and anthropogenic factors explained a greater proportion of the SDD variance, with NTL (13.84%) and fertilizer (11.17%) being the two key factors (Figure 5b and Figure S9b). As noted previously, the greater proportion of variance explained by these two variables may be due to the finer scale of analysis (compared to the provincial scale), making it possible to capture the effect of even small disturbances. As shown in Figure 5b, wastewater was another dominant factor in some watersheds, particularly for the watersheds located in the Yangtze Plain, North China Plain, and East China. This is reasonable since secondary industrial production and major population centers are mainly distributed in these regions. In addition, in some watersheds, temperature played an important role in explaining SDD variance. Since most lakes in China are not *n* or *p*-limited [9], an increase in temperature could result in increased algal abundance [8], which ultimately could translate into lower SDD. Although a minor proportion of the SDD variance was explained by precipitation, its impact may be contained in the NDVI variation. Wind speed contributed to SDD variance in 7 (out of 82) watersheds. Altogether, in 56 watersheds (out of 82), anthropogenic factors explained a greater proportion of the SDD variance compared to natural factors ($31.56 \pm 16.42\%$ vs. $21.43 \pm 11.01\%$).

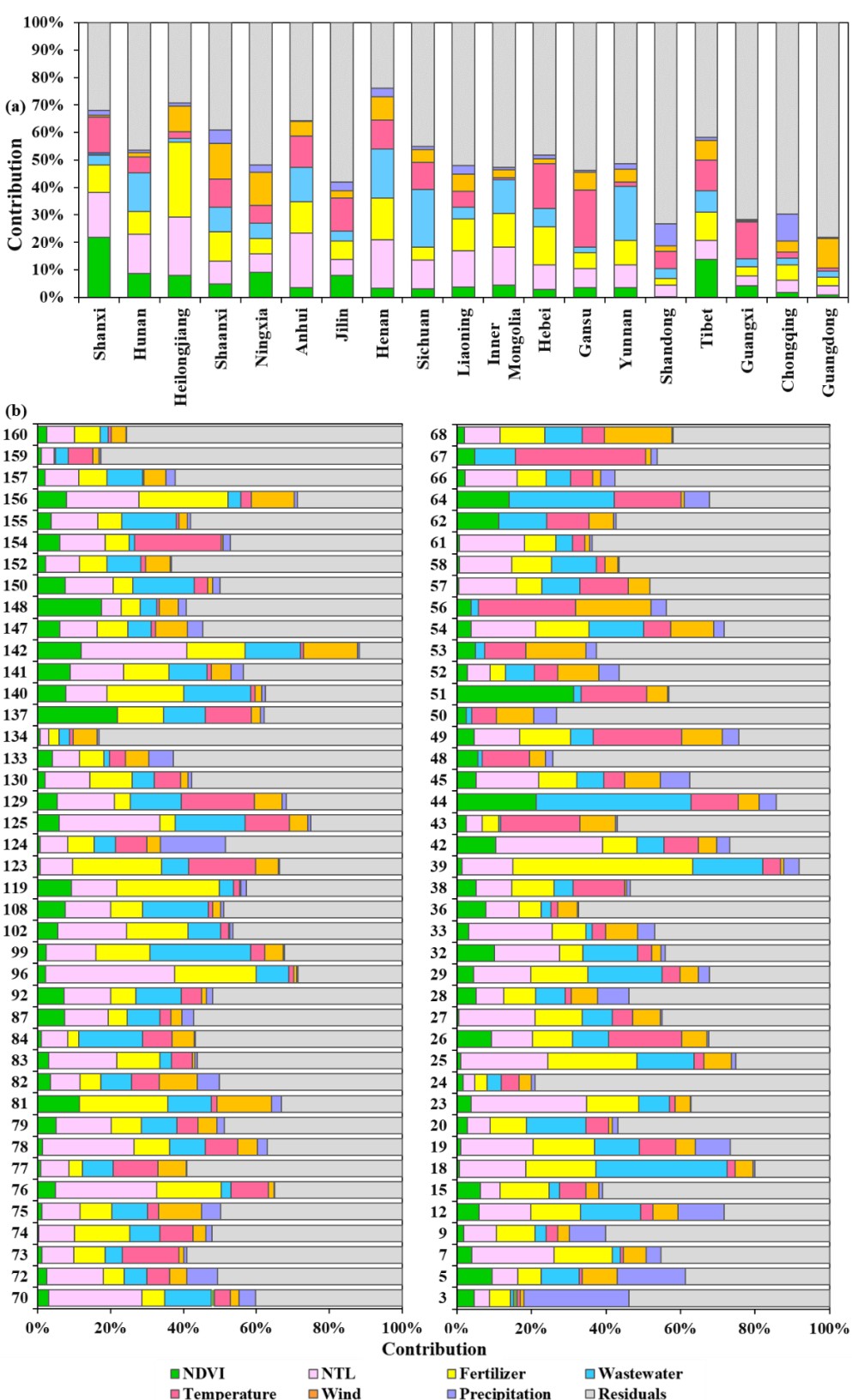

**Figure 5.** The relative contributions (in percentage) of the seven examined factors driving the interannual changes in mean SDD during 1984–2018 in the nineteen provinces (**a**) and eight-two watersheds (**b**) that had shown significant (*p* < 0.05) trends in mean SDD. General boundaries of these provinces and watersheds are depicted in Figure S2.

## 5. Conclusions

As a comprehensive indicator of water eutrophication, encompassing nutrient enrichment, algal abundance, and suspended sediment, SDD can serve as a valuable index for tracking the ecological health of aquatic ecosystems and guiding the actions of water resource managers. This information is highly valuable and can be extracted from archived satellite data. Based on the published long-term SDD datasets (1984–2018) derived from Landsat observation, the results of spatio-temporal variation were analyzed at multiple scales, from five lake regions to watershed scale. Lake clarity was lowest in Northeast (0.60 ± 0.09 m) and East China (1.23 ± 0.17 m) and highest in the Tibet Plateau (3.32 ± 0.38 m). Our results showed a significant increasing trend in lake SDD in 18 provinces (out of 32) in the past 35 years (decreasing trend in the Yunan province and no trend in the other provinces). At the watershed scale, an increasing and decreasing trend in SDD was observed in 77 and 5 watersheds, respectively. The results of SDD interannual change trends showed they were closely related to the spatial scale of analysis. In the watersheds where a significant trend in SDD was observed, HPCA analysis showed that NTL (13.84%), fertilizer use (11.17%), and wastewater discharge (9.64%) explained more of the variance than temperature (7.37%), wind speed (5.59%), NDVI (5.26%), and precipitation (3.22%), underscoring the overriding effect of anthropogenic driving factors on the temporal evolution of lake SDD. The analysis results of this study would help environment managers at local, provincial, or even national levels in decision-making for improving or protecting inland water bodies across China.

**Supplementary Materials:** The following supporting information can be downloaded at: https://www.mdpi.com/article/10.3390/rs14205091/s1. Figure S1: Spatial distribution and temporal trend in normalized difference vegetation index (NDVI, a) and night-time light brightness (NTL, b) across China. The displayed 19 provinces were that the lake clarity in these provinces showed a significant changing trend. Figure S2: Boundary of China provinces (a) and watersheds (b) included in this study. A number is assigned to each watershed in order to facilitate description and reporting of results. Figure S3: Total amount of fertilizer used in agriculture (a) and wastewater discharge (b) in 19 provinces of China during the period 1984–2018. Figure S4: Meteorological data across provinces (left graph panels) and watersheds (right graph panels). The red dots correspond to the location of weather stations at the province (a) and watershed (b) level. Mean annual precipitation (province: c; watershed: d), temperature (province: e; watershed: f), and 2-min wind speed (province: g; watersheds: h) for the period 1984–2018 are presented. Figure S5: The land use/land cover used in this study displayed every 5 years from the year 1980 to the year 2015 with a spatial resolution of 1km. Figure S6: Non-significant ($p > 0.05$) and significant increasing (slope > 0 and $p < 0.05$) trends in water clarity are presented for different lake regions across China during the period 1984–2018. The green boundary depicts regions of maximum increasing trends in water clarity; the bright green boundary represents areas of minimum increasing trends. Figure S7: Distribution of correlations (significant at $0 < p < 0.05$ and $p < 0.01$) between interannual mean SDD and the driving factors (natural and anthropogenic) at the province level (a and b) and watershed level (c and d). Figure S8: The contribution distribution of seven driving factors in province-level (a) and watershed-level (b), respectively. The circles with different colors represented diverse driving factors, and the largest circle demonstrated that one driving factor had the highest contribution in the relevant watershed, followed by others. Table S1: Summary description of the different data sources used in this study.

**Author Contributions:** Conceptualization, H.T. and K.S.; Methodology, H.T., K.S. and G.L.; Software, G.L. and Q.W.; Formal Analysis, H.T.; Investigation, J.H.; Resources, J.H.; Data Curation, H.T. and J.H.; Writing—Original Draft Preparation, H.T.; Writing—Review and Editing, K.S., Z.W., Y.S. and S.L.; Visualization, H.T.; Supervision, K.S.; Funding Acquisition, K.S., G.L., Z.W., S.L., Y.S. All authors have read and agreed to the published version of the manuscript.

**Funding:** This research was funded by the National Key Research and Development Project of China (2021YFD1500101), National Natural Science Foundation of China (42171374, 42171385, 42071336), China Postdoctoral Science Foundation (2020M681056), Young Scientist Group Project of Northeast Institute of Geography and Agroecology, Chinese Academy of Sciences (2022QNXZ03), Special Research Assistant Project of Chinese Academy of Sciences granted to Yingxin Shang, Central Government Guides Local Funds for Scientific and Technological Development (202002047JC), and Research instrument and equipment development project of Chinese Academy of Sciences (YJKYYQ20190044).

**Data Availability Statement:** Not applicable.

**Conflicts of Interest:** The authors declare no conflict of interest.

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
