# Peer review of "Analysis of Spatio-Temporal Dynamics of Chinese Inland Water Clarity at Multiple Spatial Scales between 1984 and 2018"

_remotesensing, doi:10.3390/rs14205091_

Round 1

Reviewer 1 Report

The manuscript has been evaluated. The subject matter fits within the journal's scope and readers' interests. The authors used the published long-term SDD datasets (1984-2018) to analyze spatiotemporal patterns and the response of climatic and socio-economic factors across China at multiple spatial scales. Such an analysis result is significant for local policy-makers to manage and protect water resources. Moreover, the comprehensive analysis of the driving factors of SDD spatiotemporal variation sounds, especially considering NTL, fertilizer use, and wastewater discharge as anthropogenic factors. 

However, there are still some major concerns: (i) The published SDD datasets should be described in a more detailed way, such as the algorithms used and the accuracy. (ii) The authors analyzed the SDD interannual variation at the scales, such as lake region, province, and watershed. Is it possible to analyze SDD variations at the individual lake level? (iii) The authors conducted a detailed analysis of interannual variation. However, there is a lack of analysis of seasonal variations. It will make this study more interesting and abundant. 

Specific comments:

1. Line 14: "…… in water clarity, and examined……" the comma should be deleted.

2. Line 40: Change "invest" into "invests".

3. Lines 82-84: It should be the "investigate the correlation analysis and relative contribution of natural and anthropogenic……".

4. The content of Sec. 2.1 could be enriched, like the climate situation in different lake regions.

5. Lings 118-131: The authors could add one or two sentences to briefly illustrate why they chose the NTL, fertilizer use, and wastewater discharge as anthropogenic factors.

6. Lines 176-178: The description of the characteristic of HPCA could be more detailed. 

7. Line 256: It should be "changing trends".

8. Water temperature data and air temperature data are different. SDD can be better explained if water temperature data are available (due to its effects on algae growth and nutrient).

Author Response

Thank you for your careful and professional comments, the detailed response could be seen in the following attachment.

Author Response

Thank you for your careful and professional comment, the detailed response could be seen in the following attachment.

Round 2

Reviewer 1 Report

Many thanks to the authors for improving the manuscript and all my concerns have been addressed.

Reviewer 2 Report

I have no further comments, it can be accepted for publication.